# Mitochondrial One-Carbon Metabolism and Alzheimer’s Disease

**DOI:** 10.3390/ijms25126302

**Published:** 2024-06-07

**Authors:** Yizhou Yu, L. Miguel Martins

**Affiliations:** MRC Toxicology Unit, University of Cambridge, Gleeson Building, Tennis Court Road, Cambridge CB2 1QR, UK

**Keywords:** Alzheimer’s disease, mitochondria, one-carbon metabolism, folate

## Abstract

Mitochondrial one-carbon metabolism provides carbon units to several pathways, including nucleic acid synthesis, mitochondrial metabolism, amino acid metabolism, and methylation reactions. Late-onset Alzheimer’s disease is the most common age-related neurodegenerative disease, characterised by impaired energy metabolism, and is potentially linked to mitochondrial bioenergetics. Here, we discuss the intersection between the molecular pathways linked to both mitochondrial one-carbon metabolism and Alzheimer’s disease. We propose that enhancing one-carbon metabolism could promote the metabolic processes that help brain cells cope with Alzheimer’s disease-related injuries. We also highlight potential therapeutic avenues to leverage one-carbon metabolism to delay Alzheimer’s disease pathology.

## 1. Introduction

Mitochondria are more than cellular powerhouses. Mitochondria act in their host cell as information-processing systems (reviewed in [1]). Mitochondria compartmentalise several biochemical pathways, including the oxidative phosphorylation (OXPHOS) system and a branch of folate-dependent one-carbon (1C) metabolism [2]. One-carbon metabolism is a process found in both prokaryotes [3] and eukaryotes [4]. In eukaryotes, 1C metabolism plays a fundamental role in metabolic pathways including nucleotide synthesis, DNA and protein methylation, and mitochondrial metabolism, which are linked to Alzheimer’s disease (AD) risk.

The cause of AD remains debated. The familial forms of AD are mostly linked to the extracellular accumulation of amyloid-β peptides and the hyperphosphorylation of tau [4,5]. In contrast, late-onset AD can be more multifactorial [6,7], with a complex genetic architecture [8] that interacts with environmental factors [9]. Patients with AD exhibit increased DNA damage [10], mitochondrial dysfunction [11,12], dysregulated amino acid metabolism [13,14], and decreased DNA methylation levels [15]. Age is one of the greatest risk factors for AD. With age, several metabolic alterations are associated with an increase in AD severity [16]. Given the importance of 1C metabolism in healthy ageing, several studies have shown that increasing the expression of genes and the bioavailability of metabolites involved in 1C metabolism could lengthen health span and delay AD pathologies [17,18,19,20,21] and provide a basis for personalised medicine. In this review, we discuss the genetic, proteomic, and metabolic alterations in AD and the extent to which they are linked to mitochondrial 1C metabolism, with a particular focus on therapeutic opportunities for upregulating and enhancing 1C metabolism to prevent or delay AD pathology (Figure 1a). We focus on the mitochondrial component of 1C metabolism, given the availability of other reviews with a broader scope [2,22,23]. Our working hypothesis is that multiple hallmarks of AD, including Aβ accumulation, hyperphosphorylation of tau, and genetic risk factors, converge on mitochondrial dysfunction [24,25,26,27]. The upregulation of mitochondrial 1C metabolism could help neurons cope with these injuries and delay AD-related pathologies (Figure 1b).

## 2. Folate 1C Metabolism: From Synthesis to Consumption

Folate is an essential metabolite in mitochondrial 1C metabolism and the generation of 1C donors. Folates are a group of molecules that share a common structure of three chemically distinct moieties: a pteridine ring, a para-aminobenzoic acid (PABA) linker, and one or more glutamates (Figure 2). 

Folate, often referred to as vitamin B9, cannot be synthesised de novo in mammals, but it can be absorbed in the intestines from gut microbes or from ingested supplements or food as part of a normal diet [28,29]. In plants, folate synthesis is also compartmentalised: pterin and PABA precursors are made in the cytosol and plastids, respectively, and the assembly of folate from these precursors occurs in the mitochondria [30]. The term folate stems from the Latin word for leaf, folium, since folates are abundant in plant leaves [31]. Although substantial efforts have been made to increase folate production in plants [32], how much folate becomes bioavailable after the consumption of these plants remains uncertain. For instance, Mitchell and coworkers first purified folate in 1941. They noted that folate is present in animal tissues, green leaves, and mushrooms but not in processed foods [31], which suggests that folates can be lost in the processing of foods. Accordingly, folates are susceptible to UV photodegradation [33], light exposure [34], oxidation [35] and an acidic environment [35]. The folate synthesised by gut bacteria might thus constitute an important source. 

Pioneering work from the Goldman group showed that cellular folate is imported by folate transporters in the plasma membrane [36] (Figure 3). Cells in the small intestine transport folate in the blood via the proton-coupled folate transporter (PCFT), coupled to the cotransport of proton ions [37]. Folate receptor α transports folate across the mammalian blood–brain barrier and choroid plexus into the brain and cerebrospinal fluids [38,39,40,41]. Autoantibodies against this receptor can block its function and have been shown to occur in psychiatric conditions [42,43,44].

Reduced folate carrier (RFC), which is found in most cell types, mediates the transport of folate into the cell [45,46,47]. In the cytoplasm, dihydrofolate reductase (DHFR) metabolises folate to tetrahydrofolate (THF) while oxidising two molecules of nicotinamide adenine dinucleotide phosphate (NADPH) to NADP^+^ [48] in a two-step process. THF participates in multiple metabolic processes such as the methylation of DNA and proteins. THF can also be imported into the mitochondria via mitochondrial folate transporters (MFTs) [49]. 

In mitochondria, serine hydroxymethyltransferase (SHMT2) catalyses the hydrolysis of THF to 5,10-methenyl-THF [50,51,52] (Figure 3). Subsequently, the NAD-dependent methylenetetrahydrofolate dehydrogenase-methenyltetrahydrofolate cyclohydrolase (Nmdmc), a bifunctional enzyme also known as mitochondrial methylenetetrahydrofolate dehydrogenase (MTHFD2/L), catalyses the formation of 10-formyl-THF from 5,10-methylene-THF by reducing either NADP^+^ or NAD^+^ to NADPH or NADH [53]. 10-Formyl-THF can have different fates. Mitochondrial 10-formyltetrahydrofolate dehydrogenase (ALDH1L2) can catalyse the oxidation of 10-formyl-THF to carbon dioxide (CO_2_) while reducing NADP^+^ to NADPH [54,55,56]. Mutations in ALDH1L2 are associated with neurological abnormalities [57] and an increase in reactive oxygen species [56]. Conversely, another methylenetetrahydrofolate dehydrogenase (MTHFD1L) directs the metabolites of mitochondrial 1C metabolism into the methyl and nucleotide biogenesis pathways by converting 10-formyl-THF to formate [58], which is transported into the cytosol to produce purines and methylate S-adenosylmethionine (SAM). The cytosolic component of 1C metabolism can potentially compensate for potential defects in the mitochondrial 1C pathway [59], suggesting some metabolic robustness. 

Therefore, the folate-dependent 1C pathway fuels the synthesis of purines, contributes to the reduction of important redox co-factors NAD(P)^+^ to NAD(P)H, and enables the synthesis of the main methylation factor, SAM. Due to the importance of folate 1C metabolism for cellular health, multiple countries, including the United States, have implemented regulations to supplement cereal grains used for wheat flour with folic acid as a strategy to increase the plasma folate levels in the human population [60].

Cells metabolise folate to generate 1C units and fuel 1C metabolism in the mitochondria. During mitochondrial 1C metabolism, dihydrofolate reductase (DHFR) reduces folate to dihydrofolate (DHF) and tetrahydrofolate (THF) while oxidising two molecules of NADPH to NADP+. Mitochondrial folate transporters (MTFs) import THF into mitochondria. Serine hydroxymethyltransferase (SHMT2), which is part of the mitochondrial glycine biosynthetic pathway, reduces THF to 5,10-methylene-THF, which is oxidised by methylenetetrahydrofolate reductase 2 (MTHFD2 or MTHFD2L) to 10-formyl-THF in a two-step process. 10-formyl-THF is used in different metabolic reactions, including the production of formates through methylenetetrahydrofolate dehydrogenase 1L (MTHFD1L), NADPH and CO_2_ through aldehyde dehydrogenase 1, L2 (ALDH1L2), and the production of methionine through mitochondrial methionyl-tRNA formyltransferase (MTFMT). One-carbon metabolism-related proteins are distributed in the nuclear, mitochondrial, and cytoplasmic compartments [61]. Only the mitochondrial compartment of the 1C metabolism is shown in Figure 3.

## 3. Fuelling the Cellular Nucleotide Pool and Promoting Mitochondrial Biogenesis by Enhancing Mitochondrial 1C Metabolism

One-carbon metabolism plays a crucial role in replenishing pools of nucleotides (both purines and pyrimidines) to maintain substrate bioavailability for DNA repair. Specifically, 10-formyl-THF can be metabolised to inosine monophosphate (IMP) for de novo purine (adenosine and guanosine) synthesis. 

Deficiencies in 1C metabolism can cause embryonic lethality or birth defects [62,63]. For example, impairment of 1C metabolism, such as through the suppression of SHMT1, causes neural tube closure defects through impaired nucleotide synthesis [64]. The replenishment of the nucleotide pool through deoxyuridine supplementation can prevent these defects [62], highlighting the role of 1C in nucleotide synthesis.

Various cellular processes in neurons, including nuclear DNA repair and the replication of mitochondrial DNA (mtDNA), rely on the availability of nucleotides [65]. Single-nucleus sequencing results from human post mortem brains showed that neurons accumulate approximately 20 genomic single-nucleotide variations per year, and neurons from patients with AD have significantly more DNA variations than those from age-matched controls [10]. During DNA repair, damaged nucleotides are removed and replaced by new nucleotides to restore the DNA sequence [66]. Oxidative damage to DNA creates oxidised DNA bases, of which deoxyguanosine oxidation to 7,8-dihydro-8-oxoguanosine (8-oxoG) is a major mutagenic event [67]. The DNA damage pattern in human neurons suggests that misrepaired damage to the deoxyguanosine nucleotide could be the cause of AD-related DNA mutations [10]. It is thus conceivable that an imbalanced nucleotide pool, such as a decreased availability of guanosine, could exacerbate the accumulation of somatic mutations in nuclear DNA through age and in AD (Figure 4). To test this hypothesis, it would be important to test the relative quantities of free nucleotides in patients with AD compared to controls, possibly using liquid chromatography-mass spectrometry (LC-MS). If there is indeed an imbalance in nucleotides, future research can investigate whether increasing nucleotide synthesis by upregulating 1C metabolism or taking deoxyguanosine as a supplement could delay AD pathology and decrease somatic mutations. 

An analysis of the intracellular pools of deoxyguanosine triphosphate indicated that this metabolite is concentrated in the mitochondrial matrix [68,69,70,71]. Deoxyguanosine kinase (DGUOK) is a mitochondrial enzyme that catalyses the first step of the phosphorylation of deoxyguanosine to its monophosphorylated form (dGMP). This first step in the phosphorylation of deoxyribonucleosides (dNs), such as deoxyguanosine, is the rate-limiting step in the salvage pathway of nucleotide synthesis [72]. Neurons are postmitotic cells that rely on the salvage biosynthetic pathway for nucleotide generation [73]. Mutations that impair DGUOK function cause mtDNA depletion syndrome [74], and a folate-mediated increase in nucleotide bioavailability through the de novo biosynthetic pathway leads to an increase in the mtDNA copy number [75]. Improving the mitochondrial function in models of AD by upregulating 1C metabolism might result in neuroprotection. Most cancer cells require an abundant nucleotide pool to sustain their high proliferation rate [76,77]. It is conceivable that in nonreplicating or postmitotic cells like neurons, the nucleotide pool could play a similar role in the replication of mtDNA and therefore sustain mitochondrial biogenesis. Mitochondria contain a streamlined genome in their circular mtDNA that encodes 13 proteins for energy production [78]. The replication of mtDNA is crucial for mitochondrial biogenesis [79], and an imbalance in the pool of nucleotides is linked to mtDNA depletion [80,81]. Additionally, a decrease in the number of mtDNA copies was observed in some patients with AD [82,83,84]. This observation supports Wallace’s hypothesis, which suggests that mtDNA quantity and quality are measures of biological ageing and degenerative diseases [85]. Patients with mutations in DGUOK show a severe depletion of their nucleotide pools and a decrease in the enzymatic activities of mitochondrial-encoded respiratory complexes, leading to death before one year of age [80]. It is likely that the severe disease observed in these patients results from the depletion of their pools of nucleotides [80,81]. There are some results suggesting that post mortem brain samples from patients with AD also have altered levels of genes involved in nucleotide metabolism [86]. Ensuring an efficient flux of deoxyribonucleotides by enhancing mitochondrial 1C metabolism could thus ensure the presence of sufficient quantities of nucleotides to fuel mtDNA replication, promote mitochondrial biogenesis, and protect against AD pathology (Figure 4). This possibility is supported by the findings of a study showing that enhancing 1C metabolism via folate supplementation increased nucleotide bioavailability in vivo, increased mitochondrial biogenesis, and decreased neurodegeneration in Parkinson’s disease models [75]. Future research is needed to systematically investigate the genes linked to nucleotide bioavailability and 1C metabolism.

mtDNA quantity and quality seem to vary across brain regions [82,87], among tissue types [88], and possibly among patients with AD. It is possible that, due to the multifaceted nature and high interindividual variability of AD pathology, only a subset of people could have insufficient mtDNA, while others have normal levels. Different cell types might have variable levels of mtDNA quality and quantity. We can also leverage advances in single-cell technologies to investigate mtDNA quantity and quality at the cellular level in the AD brain [88,89]. Taken together, future research could investigate the links among the levels of deoxynucleotides, DNA integrity, and mitochondrial biogenesis in individuals with AD and during ageing. 

## 4. Role of 1C Metabolism in the Biosynthesis of Nucleotide-Containing Cofactors

In addition to acting as a direct substrate for DNA repair or synthesis, nucleotides or nucleotide derivatives can also act as cofactors for enzymatic reactions. Deoxyguanosine triphosphate (dGTP), for instance, binds to mitochondrial complex I [68,90] and could contribute to protein stability [91]. Mitochondrial complex I oxidises the reduced form of nicotinamide adenine dinucleotide (NADH) to NAD^+^ while reducing ubiquinone to ubiquinol to transport protons across the mitochondrial inner membrane [92]. Patients with AD have lower mitochondrial complex I activity [11]; therefore, sufficient dGTP levels could improve the function of mitochondrial complex I by improving its stability in these patients. 

The defects in complex I function [11,26] and lower NAD^+^ levels in patients with AD indicate that the reduced activity of this respiratory complex could result in a decrease in NAD^+^ levels. Ageing [93], neurodegenerative diseases such as AD [12,94], and epigenetic dysregulation [95,96] are all linked to decreased NAD^+^ levels. NAD^+^ and its phosphorylated form, NADP^+^, are important cofactors in electron transfer and redox reactions. Increasing the NAD^+^ levels in cells with defects in their mitochondrial electron transport chain rescues proliferative defects [97]. The introduction of an NADH dehydrogenase to increase the NAD^+^/NADH ratio transgenically increases the lifespan of flies [98]. NAD^+^ levels are also decreased in animal models of AD [12,99,100]. The restoration of NAD^+^ levels has shown promising effects in animal models of neurodegeneration [12,99,101,102,103], and these approaches are currently being tested in clinical trials [104]. Taken together, this significant body of evidence indicates that increasing NAD^+^ metabolism could be a viable strategy for delaying AD onset or mitigating AD symptoms by improving mitochondrial function.

Feeding fly models of Parkinson’s disease associated with a loss of mitochondrial function with folic acid showed increased NAD^+^ levels, resulting in neuroprotection [75], which highlights the importance of 1C metabolism as a source of NAD^+^. Given the role of 1C metabolism in purine metabolism, boosting the mitochondrial 1C pathway may increase the pools of adenine used to make NAD^+^. In cellular and animal models, nicotinamide mononucleotide adenylyltransferase transfers the adenosine moiety of ATP to nicotinamide mononucleotide to form NAD^+^ [105,106]. NAD^+^ can also be synthesised from adenosine monophosphate [107,108]. It is thus conceivable that enhancing 1C pathway activity could replenish the NAD^+^ pool in AD. Several enzymes of the mitochondrial 1C pathway play a role in NAD^+^ homeostasis. The oxidation of formyl-THF to CO_2_ and THF by the ALDH1L2 enzyme leads to the production of mitochondrial NADPH (Figure 3). NADPH can directly support the antioxidant defence system, as glutathione reductase utilises NADPH to recycle oxidised glutathione back to its reduced form. Increasing NADPH also led to a decrease in oxidative stress and an increase in health span in mice [109]. Upon mitochondrial complex I deficiency, serine catabolism through 1C metabolism can fuel the generation of NADH [50]. The generation of NADH is mediated by mitochondrial MTHFD2 (L), which acts upstream of ALDH1L2 [50]. As the upregulation of the fly orthologue of MTHFD2(L) was shown to improve mitochondrial function and rescue neurodegeneration in fly models of Parkinson’s disease [110], it would be important to test whether its action in the upregulation of ALDH1L2 and MTHFD2(L) in fly models of AD would also result in neuroprotection. Taken together, we propose that increasing 1C metabolism could protect against AD pathologies by increasing NAD^+^ levels and boosting mitochondrial function, in the context of ageing and AD, which are both linked to a decrease in mitochondrial health [111]. 

Increasing NAD^+^ levels through the 1C pathway could also act to promote DNA repair. The levels of the DNA repair proteins poly(ADP-ribose) polymerases (PARPs) are increased in patients with AD [12,103,112]. PARPs catalyses the formation of ADP-ribose polymers from NAD^+^ to protein acceptors, and higher PARP activity has been described as important for maintaining genome stability across the life span in several model organisms [113]. Patients with AD have been shown to have increased DNA damage [114], which can activate PARPs [115,116] to maintain genome integrity. However, increased PARP activity could be detrimental to cellular function either by excessively lowering the bioavailability of NAD^+^ for other enzymes, such as sirtuins, or by lowering ATP levels and depleting cellular energy stores. An abundant pool of NAD^+^ might also be important for the ability of PARPs to protect cells from DNA damage [117]. Accordingly, a decrease in NAD^+^ levels may impair PARP function and compromise the accurate storage of cellular information stored in their DNA. The associations among NAD^+^ levels, PARP activity, genome integrity, and DNA repair processes in AD remain to be investigated. 

## 5. Role of 1C Metabolism in Epigenetic and Post-Translational Modifications of Proteins

In addition to preserving the fidelity of the genome by supplying cofactors for DNA repair enzymes, 1C metabolism also provides methyl groups for the methylation of nuclear DNA and mtDNA. Methylation, as an epigenetic modification, can regulate gene expression and maintain cellular identity. One-carbon metabolism provides the methyl groups necessary for DNA methylation reactions through the synthesis of SAM, a major donor of methyl groups. The levels of SAM are decreased in the brains of patients with AD [118]. SAM is generated from methionine, an amino acid obtained from dietary sources, and ATP through the mitochondrial 1C pathway [119]. Methionine adenosyltransferase, an enzyme involved in 1C metabolism, catalyses the conversion of methionine to SAM, facilitating the transfer of methyl groups to DNA and other substrates. Changes in DNA methylation are linked to lifespan and ageing [120,121]. The importance of preserving accurate DNA methylation patterns is important in the contexts of AD [122,123] and ageing, as defects in this process can contribute to disease onset and age-related functional decline. Reprogramming somatic cells into pluripotent stem cells by expressing Yamanaka factors also requires 1C metabolism to maintain efficiency [124,125]. It is therefore conceivable that increasing methyl donor or genetic variations in 1C genes could be linked to DNA methylation. Clinical studies showed that genetic mutations that impair the function of MTHFR, a gene involved in 1C metabolism, are linked to lower levels of DNA methylation [126]. Dietary supplementation with folic acid is linked to higher levels of DNA methylation [126,127]. Ageing [120,128] and AD [123,129,130] are linked to lower levels of DNA methylation globally. Dietary supplementation with SAM in mouse models of AD rescued AD pathologies [131,132], indicating a causal link between SAM levels and AD pathology. The observation that feeding SAM orally in a mouse model of AD increased methylation in the brain suggests that SAM can cross the blood–brain barrier [131] and thus be used as a dietary supplement for patients with AD. However, detailed pharmacokinetic measurements of SAM in animal models and patients with AD would be required to determine whether the dietary intake of SAM could increase the levels of bioactive SAM in the brain.

mtDNA has low or no methylation [133]. However, some studies have reported that mtDNA can be methylated by DNA methyltransferases (DNMTs) [134,135,136]. The levels of mtDNA methylation are altered in both animal models of AD [137] and some patients with AD [15,122,138,139,140]. The D-loop region of the mitochondrial genome regulates the replication of mtDNA and its transcription [141,142,143,144] and is methylated in both model animals and humans [145,146]. Both the hypo- and hypermethylation of the D loop have been observed in AD [138,140,147]. This variation in the degree of methylation might be explained by variations in disease stage or the brain area used for the analysis. DNMT3a activity can alter the mtDNA 5-methylcytosine levels [148], both of which are changed in AD [149]. The levels of 5-hydroxymethylcytosine in the mtDNA were decreased in aged animals compared to their younger counterparts, in line with changes in the mRNA levels of the enzymes involved in this methylation [150]. The functional consequences linked to the methylation of mtDNA patterns, including D-loop methylation, on mitochondrial dynamics remain underinvestigated. The overexpression of the DNA methyltransferase 1 variant with a mitochondrially targeted sequence (mtDNMT1) led to the specific upregulation of ND1 mRNA levels [136]. ND1, a subunit of mitochondrial complex I, is essential for initiating complex I assembly [151]. It is possible that altering mtDNA methylation through mtDNMT1 could regulate mitochondrial gene expression and rescue mitochondrial dysfunction in AD.

In addition to the methylation of mtDNA, the methylation status of the NADH dehydrogenase 5 RNA transcript is elevated in cells treated with Aβ and patients with AD. This increased methylation leads to translational repression and mitochondrial dysfunction in AD [152]. A global investigation of N6-methyladenosine levels in a mouse model of AD showed that RNA methylation is elevated in the mouse model [153]. Consistently, enzymes involved in RNA methylation are also differentially expressed in AD [154,155]. Interestingly, mechanistic target of rapamycin complex 1 (mTORC1), which has been linked to AD risk and longevity, is also a key regulator of RNA modification through the addition of N6-methyladenosine [156,157,158,159]. mTORC1 introduces an N6-methyladenosine modification to MXD2, which results in its degradation [160,161]. MXD2 is a negative regulator of cMYC, and its degradation causes an increase in cMYC activity and a global upregulation of translation [160,161]. Endoplasmic reticulum stress, which is associated with translational shutdown, is prevalent in individuals with AD [162]. Increasing mTORC1-dependent translational upregulation could thus be a potential avenue to reverse translational stress in AD.

Protein methylation is a type of post-translational modification that can play an important role in regulating protein function. The bioavailability of the 1C donor SAM is critical to mitochondrial function as it enables mitochondrial complex I assembly and iron-sulphur cluster biosynthesis [163]. Arginine methyltransferase NDUFAF7 methylates arginine 85 of NDUFS2, a complex I subunit that facilitates the assembly of complex I [164,165] in a SAM-dependent manner [163]. Impairment of complex I activity can lead to ROS production and neurodegeneration, and patients with AD have impaired complex I activity in the brain [11,26]. Facilitating complex I assembly through 1C metabolism could potentially alleviate AD progression. More widely, the methylation of other proteins involved in AD could also be linked to neuronal function. For instance, the microtubule-associated protein tau can undergo extensive post-translational modifications. Tau phosphorylation increases its propensity to aggregate [166]. On the other hand, methylation of tau has been detected in the brains of patients with AD [167], and biochemical analyses have indicated that increased methylation decreases the propensity of tau to aggregate and protects against its neurotoxic effects [168]. Future work could leverage advances in mass spectrometry to unbiasedly examine the difference in the protein methylome between patients with AD and controls [169]. 

## 6. Therapeutic Opportunities and Novel Methods of Enhancing Mitochondrial 1C Metabolism

Based on previous observations that increasing mitochondrial 1C metabolism could delay AD phenotypes at multiple omic levels, research on the effect of supplementation with the precursor metabolite folate on AD pathology has shown encouraging results. Here, we discuss various results and limitations on the potential of using folate as a therapy for AD, focusing on clinical data. 

The reference range for the concentration of folate in plasma or serum in humans typically spans from 13.5 to 45.3 nM [170,171]. Despite several countries adopting regulations on folic acid fortification [172], patients with AD have lower folate levels in their plasma than controls [18]. Dietary supplementation with more than 400 μg of folate per day in patients with AD slowed their cognitive decline [173,174,175]. There are, however, numerous results that show a lack of association between folate supplementation and the suppression of AD pathology [176,177,178], suggesting some potentially heterogeneous effects of folate supplementation. 

Understanding these discrepancies is key for evaluating the therapeutic potential of increasing mitochondrial 1C metabolism to delay AD pathology. A potential issue with studies investigating the link between folate intake and subsequent AD risk using epidemiological methods [176,177] would be reverse causation: it is possible that individuals with increased frailty or AD risk could be self-medicating by taking folic acid supplements. Randomised clinical trials to test the efficacy of dietary supplementation with folates compared to a placebo would address this shortcoming. However, in a clinical trial of the effect of folic acid supplementation on individuals with mild cognitive impairments, it was reported that the effect of folate supplementation on improving cognition was not statistically significant [179]. Another putative explanation for the lack of efficacy of folate dietary intake might result from individual differences in the capacity to absorb and metabolise folate. Several proteins, including folate receptors, reduce folate carriers and proton-coupled folate transporters [180,181,182,183] and regulate the transport of folate in different tissues, including the blood–brain barrier and the intestine. Differences in the expression levels of these folate transporters might exist in patients with AD, which could alter the efficiency of folate transport across the blood–brain barrier in individual patients. For instance, the vitamin D receptor regulates the expression of folate transporters, and treating mice with the active form of vitamin D increased the transport of folate in the brain [184]. Previous research showed that patients with AD have lower vitamin D levels than healthy controls [185], which could result in the lower expression of folate transporters. Characterising the expression of folate transporters in patients with AD could inform whether taking folate orally would be sufficient to increase mitochondrial 1C metabolism in the brain. Since vitamin D might play a role in the transport of folate, it would be informative to reanalyse the epidemiological data on folate intake, while accounting for serum vitamin D concentrations. The hypothesis would thus be that folate and vitamin D combined supplementation could have an increased beneficial effect on preserving cognitive health in patients with AD compared to the effects of folate supplementation alone.

Folate is metabolised intracellularly into more biologically active molecules such as THF or SAM through mitochondrial 1C metabolism. The transcripts encoding for the components of 1C metabolism such as *DHFR* are altered post mortem [186]. Additionally, post mortem AD brains have increased signs of an unfolded protein response [187], suggesting a decreased level of translation [188]. It is thus possible that in some patients with AD, defects in 1C metabolic pathways prevent the efficient metabolism of folate. Enhancing mitochondrial 1C metabolism directly with the supplementation of the 1C donor THF or other forms of folates could be a viable strategy to bypass such defects.

Another potential approach to increase folate delivery in patients with AD is the use of gut bacteria that produce a range of different folate products including the more metabolically ready DHF and THF [29,189]. Pioneering studies on the identification and function of folates have mostly relied on bacterial sources. Folate derived from yeast was used to treat tropical macrocytic anaemia in humans [190,191]. Folate was also first detected in *Streptococcus lactis* [31]. These observations suggest that microbes containing the metabolic pathways involved in folate synthesis can excrete forms of folate that are more metabolically active to be absorbed by the gut. An important question remains as to whether the microbes that contain genes for de novo folate synthesis still produce folate in the human gut. The caecum and colon of rats fed human milk containing Bifidobacterium, a folate producer, were engrafted with this bacterial strain. This resulted in increased plasma folate levels in these animals [192]. In humans, the supplementation with Bifidobacterium (*Bifidobacterium adolescentis* DSM 18350, *Bifidobacterium adolescentis* DSM 18352, and *Bifidobacterium pseudocatenulatum* DSM 18353) was linked to increases in folate levels and faecal Bifidobacterium count. We suggest that manipulating the gut microbiome and feeding animal models with bacteria that produce different forms of folates could be a viable alternative to dietary folate intake to enhance mitochondrial 1C metabolism in the host.

Taken together, upregulating mitochondrial 1C metabolism through pharmacological and probiotic methods could be beneficial for patients with AD. We envisage that patients could adopt these treatments in addition to existing AD treatments such as cholinesterase inhibitors (reviewed in [193]) and antibody-based therapies [194].

## 7. Conclusions and Outlook

Here, we focused on the potential mechanistic links between mitochondrial 1C metabolism and AD. We propose that upregulating 1C metabolism is a potential therapeutic approach for countering the neurodegeneration in AD by improving mitochondrial health. Mitochondrial 1C plays vital roles in cellular and organismal functions. These roles range from maintaining mitochondrial function to ensuring normal foetal development in vertebrates. We reason that leveraging the increasing amount of multimodal data, such as genomic, transcriptomic, methylomic, proteomic, and metabolomic data, from different human tissues, to explore novel approaches to increase 1C metabolism could result in new approaches to promote healthy ageing. These insights could also elucidate the interindividual differences in mitochondrial 1C metabolism, which could inform the design of personalised treatments for neurodegenerative diseases such as AD.

## 8. Perspectives

Mitochondrial 1C metabolism plays a central role in various biochemical processes, including mitochondrial function, nucleotide synthesis, amino acid metabolism, and epigenetic regulation. These processes are intertwined with the mechanisms of ageing and the onset of AD.Mitochondrial 1C reactions produce 1C units while synthesising coenzymes such as NADPH. Mitochondrial 1C metabolism differs in cell-type and disease-specific contexts. Defects in this metabolic pathway lead to multiple repercussions, such as imbalances in crucial metabolites, oxidative stress, and defects in DNA repair.Further research on the causal effects of genes and metabolites in the 1C pathway could elucidate healthy ageing trajectories and the mechanisms of AD.

## Figures and Tables

**Figure 1 ijms-25-06302-f001:**
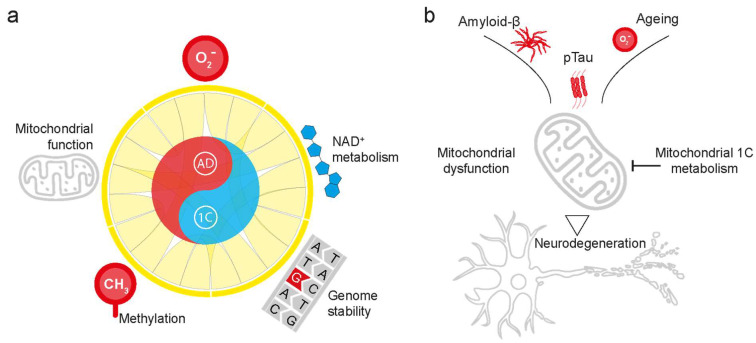
Links between mitochondrial 1C metabolism and AD. (**a**) An illustration of the 5 general cellular processes linked to both Alzheimer’s disease (AD) (in red) and 1C metabolism (in blue), which are the main topics of this review (in grey): redox metabolism, NAD^+^ metabolism, mitochondrial function, genome stability, and the methylation of DNA and proteins have been shown to be dysregulated in AD. (**b**) Working hypothesis on the potential of upregulating mitochondrial 1C metabolism to delay neurodegeneration in AD. AD-related pathologies such as the accumulation of toxic amyloid-β, hyperphosphorylated tau (pTau, and age-related accumulation of oxidative damage can lead to mitochondrial dysfunction and neurodegeneration. Upregulating mitochondrial 1C metabolism has the potential to alleviate AD-related damage and aid cells in coping with neurotoxic damage.

**Figure 2 ijms-25-06302-f002:**
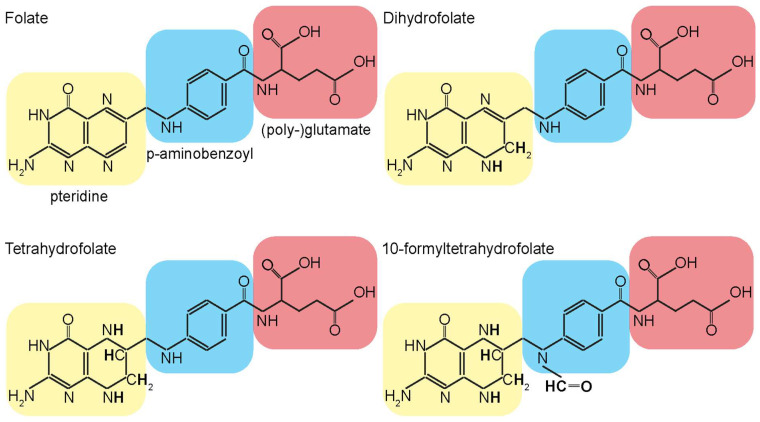
Structures of folate molecules. Folate molecules are composed of three distinct chemical components. At the core is a pterin heterocyclic ring (2-amino-4-hydroxy-pteridine, yellow), which is connected via a methylene bridge to a p-aminobenzoyl group (blue). This group is further linked through an amide bond to either a single glutamic acid molecule or a chain of glutamate residues (red). Folate molecules can carry one-carbon units in various oxidation states, which may be attached to the N_5_ nitrogen atom of the pterin ring (see dihydrofolate and tetrahydrofolate for example) or the N_10_ nitrogen atom of the p-aminobenzoyl group (see 10-formyltetrahydrofolate for example). Different forms of folate molecules are shown, and the additional atoms compared to folate are in bold.

**Figure 3 ijms-25-06302-f003:**
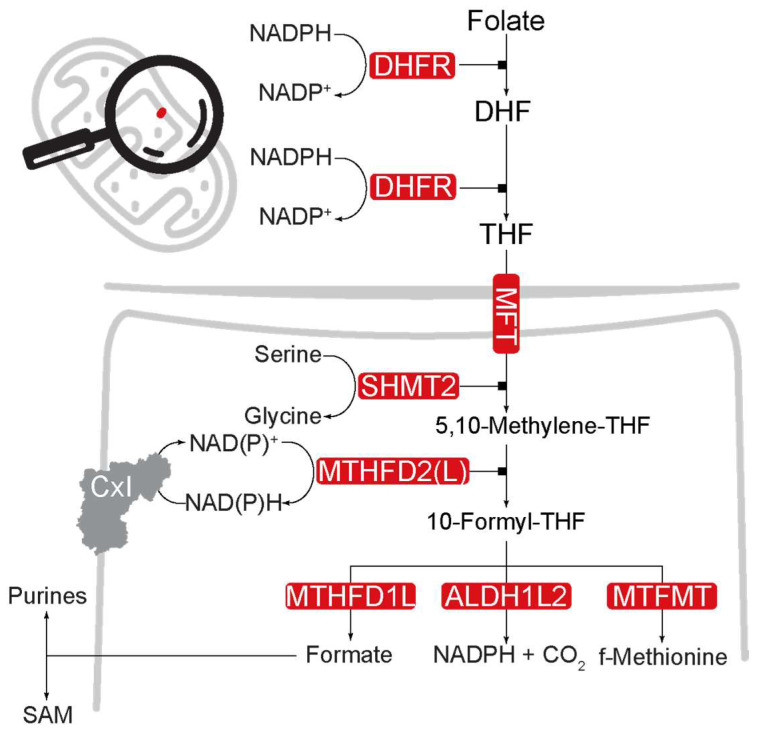
Pathway of mitochondrial 1C metabolism.

**Figure 4 ijms-25-06302-f004:**
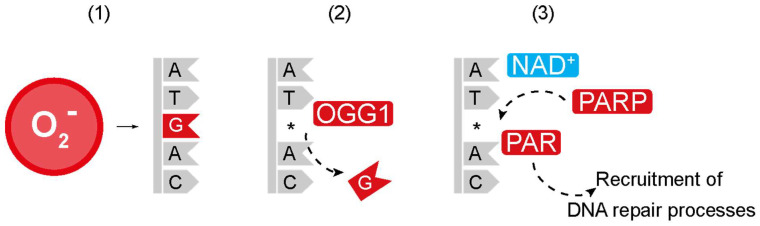
Links between mitochondrial 1C metabolism and DNA repair. Cellular stress can be caused by the accumulation of reactive oxygen species (ROS). High levels of ROS can damage DNA by converting deoxyguanosine to 8-oxo-2′-deoxyguanosine (8-oxo-dG) (1). 8-Oxoguanine glycosylase recognises and excises 8-oxo-dG, creating an apurinic site (AP site, indicated by an asterisk (2). The site where the nucleotide is lost is marked by an asterisk. Subsequently, poly(ADP-ribose) polymerases (PARPs) recognise the single-strand breaks produced from these excision processes (3). These enzymes catalyse the synthesis of poly(ADP-ribose) from NAD^+^ using ATP at the site of DNA damage. Poly(ADP-ribose) polymers (PARs) subsequently act to recruit downstream DNA repair proteins.

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
