# Peer review of "Mitochondrial One-Carbon Metabolism and Alzheimer’s Disease"

_ijms, 2024, doi:10.3390/ijms25126302_

Round 1

Reviewer 1 Report

Comments and Suggestions for Authors

This manuscript presents the relations between mitochondrial 1C metabolism and Alzheimer’s disease. The authors have reviews numerous previously published papers; however, more recent studies should be covered in this review article. Therefore, this manuscript needs major revision prior to publication. Detailed comments are provided below.

Comments:

1.        Although this manuscript cites 186 references, only 56 references (ca. 30%) have been published in the last 5 years (from 2019 to 2024). the authors should focus on the latest findings and provide a comprehensive review of the most recent researches with a suggestion of new perspective for future research.

2.        The relations between mitochondrial 1C metabolism and AD pathology should be covered in more detail.

3.        In Figure 1 and 2, the authors presented the chemical structure of folate and mitochondrial 1C metabolism. In either one of these figures, provide the chemical structural changes of folate; for example, the chemical structure of DHF, THF, 5,10-methenyl-THF, 10-formyl-THF, formate, f-methionine.

4.        The authors suggested a strategy to treat AD by improving mitochondrial health. Please provide some examples as potent tactics to enhance mitochondrial 1C metabolism to treat the disease.

5.        In addition, as AD pathology is complicated, it would be better to discuss the relations between 1C metabolism and other risk factors of AD, such as aggregation of Amyloid beta, pTau, oxidative stress.

Reviewer 2 Report

Comments and Suggestions for Authors

This is a well written review article that is timely for the AD field. The figures are appropriate and help illustrate key points. The authors provide significant background information for 1C metabolism and key pathways while also discussing alterations in the context of AD. 

Reviewer 3 Report

Comments and Suggestions for Authors

The topic is very relevant, the authors proposing that increasing one-carbon metabolism could promote metabolic processes that help brain cells cope with Alzheimer’s disease-related injuries. They also highlight potential therapeutic avenues to leverage one-carbon metabolism to delay Alzheimer’s disease pathology.

The methodology is very modern, authors using numerous metabolic pathways to demonstrate the role of 1C Metabolism in epigenetic and post translational modifications of proteins involved in Alzheimer’s disease

Finally the authors discuss therapeutic opportunities and novel methods to increase mitochondrial 1C metabolism. These results may open new research directions in the design of personalised treatments for neurodegenerative diseases such as AD.

The conclusions are consistent with the evidence and arguments presented.

The references are very relevant.

I suggest some minor editing corrections

1.       Fig. 3 is not mentioned in the text. It should be moved after the text reffering to it (row 126)

2.       Fig. 4 is not mentioned in the text. It should be moved after the text reffering to it (row 247)

3.       All the references should be written according to Authors Guide for MDPI journals (see ref 185)

4.       For section 6 you may see also

Stanciu, G.D.; Luca, A.; Rusu, R.N.; Bild, V.; Beschea Chiriac, S.I.; Solcan, C.; Bild, W.; Ababei, D.C. Alzheimer’s Disease Pharmacotherapy in Relation to Cholinergic System Involvement. Biomolecules 202010, 40. https://doi.org/10.3390/biom10010040

Round 2

Reviewer 1 Report

Comments and Suggestions for Authors

Although the authors revised the manuscript to improve its quality, however, still some points should be fixed.

1. what are the sentences from line 50? Are those the part of Figure 1 caption?

2. Please fix grammatical errors and punctuation errors. The sentences should have period. 

3. The writing should be fixed. For example, the line 141-142.

Comments on the Quality of English Language

Please fix grammatical errors and punctuation errors. The sentences should have period. 

The writing should be fixed. For example, the line 141-142.
